# Microbiome in Chronic Kidney Disease (CKD): An Omics Perspective

**DOI:** 10.3390/toxins14030176

**Published:** 2022-02-26

**Authors:** Sonnal Lohia, Antonia Vlahou, Jerome Zoidakis

**Affiliations:** 1Center of Systems Biology, Biomedical Research Foundation of the Academy of Athens, 11527 Athens, Greece; slohia@bioacademy.gr (S.L.); vlahoua@bioacademy.gr (A.V.); 2Institute for Molecular Cardiovascular Research, RWTH Aachen University Hospital, 52074 Aachen, Germany

**Keywords:** gut microbiota, CKD, uremic toxins, omics, therapeutic targets

## Abstract

Chronic kidney disease (CKD) is predominant in 10% of the world’s adult population, and is increasingly considered a silent epidemic. Gut microbiota plays an essential role in maintaining host energy homeostasis and gut epithelial integrity. Alterations in gut microbiota composition, functions and, specifically, production of metabolites causing uremic toxicity are often associated with CKD onset and progression. Here, we present the latest omics (transcriptomics, proteomics and metabolomics) studies that explore the connection between CKD and gut microbiome. A review of the available literature using PubMed was performed using the keywords “microb*”, “kidney”, “proteom”, “metabolom” and “transcript” for the last 10 years, yielding a total of 155 publications. Following selection of the relevant studies (focusing on microbiome in CKD), a predominance of metabolomics (*n* = 12) over transcriptomics (*n* = 1) and proteomics (*n* = 6) analyses was observed. A consensus arises supporting the idea that the uremic toxins produced in the gut cause oxidative stress, inflammation and fibrosis in the kidney leading to CKD. Collectively, findings include an observed enrichment of *Eggerthella lenta, Enterobacteriaceae* and *Clostridium* spp., and a depletion in *Bacteroides eggerthii, Roseburia faecis* and *Prevotella* spp. occurring in CKD models. Bacterial species involved in butyrate production, indole synthesis and mucin degradation were also related to CKD. Consequently, strong links between CKD and gut microbial dysbiosis suggest potential therapeutic strategies to prevent CKD progression and portray the gut as a promising therapeutic target.

## 1. Introduction

In 400 BC, Hippocrates distinctively stated that “All diseases begin in the gut” [1]. The gut microbiota comprises of approximately 10–100 trillion microorganisms [2,3] living in the gut of a human, in a symbiotic relationship [4,5]. These microorganisms include bacteria, viruses, fungi, archaea and unicellular eukaryotes that are composed of 3.3 million genes [6]. In human adults, the gut microbiota (frequently also referred to as ‘gut microbiome’) includes mainly the bacterial phyla *Firmicutes* and *Bacteroidetes*, dominating over 50 other phyla. The gut microbiota can also be characterized based upon its functional diversity, related to its impact on systemic immunity and host protection from enteral pathogens [1,6]. The microbial metabolism involves proteins, lipids, bile acids, vitamin synthesis and fermentation of carbohydrates [7,8]. Several factors such as age [9], drugs, exposure to allergens, hygiene and various infectious diseases [10] determine the microbiota’s diversity, uniformity and richness [11,12].

Along these lines, it is well established by now that a microbial imbalance may lead to disease development [13] via multiple mechanisms, i.e., the gut microbiota plays an essential part, among others, in maintaining the host energy homeostasis and gut epithelial integrity [7]. In addition, the gut microbiota interacts with the cardiovascular, endocrine, renal and nervous systems, and respective molecular pathways that can cause hypertension, diabetes, proteinuria and kidney and inflammatory diseases [14]. As such, understanding the host-gut microbiota relationship has become a major research goal. In this review, we focus on the molecular studies investigating the gut-kidney axis with an emphasis on chronic kidney disease (CKD). 

Ten percent of the world’s adult population suffers from CKD, and with the ever-increasing aging population, it is considered a rising global threat: a silent epidemic [15,16]. CKD is typically associated with other diseases like obesity [17], diabetes [18], hypertension [19], inflammatory bowel disease [20], proteinuria [21] and cardiovascular diseases (CVD) [22]. CKD diagnosis is defined based on either: (i)symptoms of a kidney injury for more than 3 months at normal glomerular filtration rate (GFR),(ii)an increase in the urinary albumin:creatinine ratio (UACR) above 30 mg/g,(iii)a decrease in the estimated GFR (eGFR) below 60 mL/min/1.73 m^2^ [23,24].

The exact values of eGFR and UACR can to some extent reflect the progression of CKD to end-stage renal disease (ESRD) and risk of death [25]. In addition, the strong links of gut microbiome and CKD have been observed for quite some time now. As examples: (i)In 1978, researchers observed a significant alteration in gut microbiota in CKD patients in comparison to controls, and demonstrated microbiota’s ability to metabolize substrates causing uremic toxicity [26].(ii)Later on, another study indicated significant altered abundance of 175 bacterial operational taxonomic units (OTUs) in CKD and control rats [27]. An increase in *Enterobacteriaceae/Proteobacteria* and *Bacteroidetes* spp. with concomitant decrease in *Prevotella, Bifidobacteriaceae* and *Lactobacilli* spp. were suggested to intensify the sympathetic outflow causing hypertension and CKD. We may, therefore, hypothesize an impact of this microbiome on the renin-angiotensin-aldosterone-system (RAAS) that regulates sodium and water absorption in the kidney, thus directly controlling systemic blood pressure [28].

It is generally considered that CKD causes an interference in the symbiotic relationship of host organs and gut microbiota, and in turn this gut dysbiosis participates in CKD progression [13], illustrated in Figure 1. Increased levels of uremic toxins in the kidney, gastrointestinal tract mechanical changes and colonic transit alterations are commonly observed in CKD [11,29]. In parallel, high concentrations of uric acid in the gut, due to decreased excretion by the kidney, increases urease activity [27], consequently increasing the gut pH, systemic inflammation and, collectively, leading to an imbalance of the gut microbiota composition [27,30] with an increase in abnormal metabolites and intestinal permeability [6]. A histological study performed on the intestines of CKD patients showed reduced villous height, crypts elongation and inflammatory cells infiltrating the lamina propria [31]. Therefore, the gut and kidney are involved in a complex bidirectional interaction. In addition, in the past decade scientists have proposed the integral role of intestinal dysbiosis and liver damage contributing further to renal injury, associated with an increased presence of bile acids in CKD models [32]. The accumulation of uremic toxins in the circulation due to renal injury affects gut microbiota composition and metabolites; altered microbial metabolism results in an additional increase of the major uremic toxins, furthering the renal injury in the form of a vicious cycle [33]. 

High disease incidence, and complexity of the underlying molecular events, has prompted extensive research efforts towards a better understanding of the gut-kidney axis. Application of -omics approaches, providing large scale analysis of the molecules (DNA, RNA, proteins, metabolites) in a biological system, in the field of nephrology has been on the rise for the last few years, since they are able to reflect the system complexity [34]. Omics-based studies aiming at understanding the host-gut-kidney axis are portrayed in this review. Specifically, to get an overview of the research in this area and results achieved by now, we performed a thorough review of the literature from the last 10 years (2011–2021) separately for proteomics, transcriptomics and metabolomics. The main keywords used for proteomics studies included “microb*”, “proteom*”, “kidney” and “gut”; for transcriptomics the respective keywords were “gut”, “microb*”, “transcript*” and “kidney”; whereas for metabolomics studies these included “microb*”, “CKD” and “metabolom*” (it should be noted that additional keywords were tested in each case, such as instead of “kidney”—“renal” or instead of “microb*”—“gut” etc., with overlapping results). A total of 155 publications (33 for proteomics, 78 for transcriptomics, and 44 for metabolomics) were retrieved (on 15 June 2021). After detailed analysis by at least two co-authors, 42 papers were found to be relevant to CKD, gut microbiome or omics, of which 19 were found to focus on the gut microbiome-kidney interactions targeting their better understanding, this being the main point of attention of this review. Studies focusing on a particular dietary modification for CKD patients or specific signalling pathways, pre-/pro-/syn-biotic supplement studies, non-microbial omics studies, review papers, clinical trial reports and project and conference abstracts were excluded, as represented in Figure 2. 

This manuscript, thereby, aims to provide an overview of -omics approaches as applied to the investigation of the gut-kidney axis in the context of CKD, illustrated in Figure 3. As may be observed by the number of publications, this is an active, yet early-stage, area of research. 

## 2. Transcriptomics in the Gut-Kidney-Axis

Even though, as shown (Figure 2), multiple studies were retrieved using the keyword “transcriptom*”, the vast majority was confined to the sequencing of the microbial species, without links to kidney disease and hence had to be excluded. Snelson et al. [35] investigated how long-term consumption of a processed diet drives intestinal barrier permeability and an increased risk of CKD in animal models. The researchers studied the advanced glycation (AGE) pathway, which generates Maillard reaction products within food upon thermal processing. A transcriptomic analysis was performed on diabetic mice fed with control starch (C), thermally treated (HT) control starch or with RS (HT-RS). It was discovered that on consumption of AGEs as a component of the HT processed food diet, the intestinal permeability increases. It was also observed that AGE induced the activation of the systemic innate immune complement system within the host, responsible for inflammation and kidney injury. The transcriptomic analysis indicated that on chronic exposure of heat-treated diets, the end effector molecules C3a and C5a of the proinflammatory complement system increased in HT samples in comparison to the C samples, similar to what is observed in a CKD model [36]. Inflammation and fibrosis are induced on the binding of effector molecule C5a to its receptor C5aR1 in the blood stream, accelerating renal injury. 

Meanwhile, the presence of alagebrium in the processed diet (HT+RS) proved efficient in inhibiting this AGE pathway and reversing the phenotype of kidney injury, while also reducing the activation of the complement system. Based on these results, the authors concluded that a high RS fiber diet can improve gut barrier integrity and reduce the risk of kidney injury by suppressing complement system activation. The authors also conclude that a gut targeted dietary supplement (RS) could intervene in the deteriorating actions of processed food, therefore delivering a practical way, based on improved formulation of food products, to increase a healthy life. The main findings from this study are summarized in Table 1.

## 3. Proteomics in the Investigation of Gut-Kidney-Axis

### 3.1. Dietary Modifications to Improve Gut Health in CKD

CKD patients are often prescribed low-fibre diets and limit their potassium intake to prevent cardiac arrhythmias [37,38]. However, decreased fibre intake increases gut microbial population with an impact on mucus layer and gut permeability, thus disrupting gut health and function [39,40,41]. Zybailov and co-workers [42] performed proteomic studies to investigate the modulation of host-microbiome interactions in CKD by resistant starch (RS) and digestible starch (DS) in rat models (fed with 0.7% adenine) of CKD. The reduced presence of proteins such as thioredoxin and calcylin (S100-A6) in CKD-RS, versus CKD-DS, suggested the positive effect of RS on oxidative stress and inflammation. In brief, bioinformatics (Blast2GO) analysis predicted two molecular functions activated in CKD-DS, serine-type endopeptidase inhibitor and metalloexopeptidase activities, that indicate adverse proteolytic events associated with CKD-DS. The study also reported differences in some species of *Ruminococcus* by RS versus DS supplementation, i.e., fibrolytic *Ruminoccocus* were found to be reduced with dietary RS, while amylolytic species increased; these are typically associated with butyrate producers proving to be beneficial in ameliorating oxidation and inflammation and, thus, decelerating CKD progression [43]. In conclusion, a simple change in nutrition (RS) may lead to modification of the gut microbiome that affects proteins and biological processes in the gut with potential relevance to CKD progression. 

In a follow-up proteomic study by Karaduta et al. [44] using 5/6 nephrectomy mice models fed with RS, bioinformatics analysis of the results predicted a reduction of indole synthesis in CKD-RS compared to CKD mice. This could be linked to an increased presence of butyrate-producing bacteria and decreased presence of mucin-degrading bacteria in the CKD-RS group. In addition, the initial hypothesis of improved renal function upon RS administration [42] was supported by kidney histopathology, reduced levels of harmful bacterial metabolites (such as indoxyl sulfate (IS)) and increased proliferation of beneficial bacterial species.

The physiological functions of hydrogen sulfide (H_2_S) involve post translational modifications such as S-sulfhydration of bacterial proteins (tryptophanase) and results in reduction of the levels of uremic toxins, such as IS [45,46]. It is well known that an increase in dietary protein increases the gut microbial production of these toxins [47,48], even though the mechanisms involved are not adequately described yet. A study to further determine the role of diet in altering the gut proteome (microbial and mouse) and its impact on kidney failure was performed by Lobel et al. [49] using mice models on different diets (low and high Saa (sulphur containing amino acids) +Ade (adenine)) and gut composition (germ-free (GF) vs specific pathogen-free (SPF)). At low Saa+Ade diet, the abundance of Osteopontin (Spp1), transforming growth factor beta-1 proprotein (Tgfb1) and intercellular adhesion molecule 1 (Icam1) increased in GF mice compared to SPF, whereas conversely, C-C motif chemokine 2 (Ccl2) and metalloproteinase inhibitor 1 (Timp1) increased in SPF versus GF mice. This analysis suggested the role of gut microbiota in buffering the expression of genes and intensifying CKD progression in animal models administered with low Saa+Ade diet. Along these lines, high abundance of *Escherichia coli (E. coli)* in a high Saa+Ade diet was linked to reduced levels of indole and an overall improvement of kidney function. Moreover, an in vitro experiment using *E. coli* cells (lysates that produce less H_2_S (*ΔdecR*) compared to wild types (WT)) identified 212 high confidence S-sulfhydrated proteins. Further bioinformatics analysis predicted 13 S-sulfhydrated enrichment cellular pathways, mainly associated with protein translation, and with an overrepresentation of tryptophanase (TnaA) activity. Collectively, the authors concluded that a high sulphur containing diet can induce posttranslational modification of microbially produced TnaA, reducing its activity in the gut, which in turn improves the CKD condition without changing the composition of the gut microbiome. A more detailed characterization of this molecular impact of gut TnaA onto the kidney tissue is pending.

### 3.2. Proteomics in the Heart-Gut-Kidney Axis 

Opdebeeck et al. [50] performed a detailed proteomic study to better understand the mechanism by which the uremic toxins IS and p-cresyl sulfate (pCS), produced by the gut-microbiome, promote vascular calcification in aorta and peripheral vessels of animal models for CKD, leading to cardiorenal syndrome. The key findings of this proteomic and validation study are mentioned in Table 1. An increase in tissue calcification, specifically in the medial layer of aortic wall of the CKD rats (Wistar rats under adenine and phosphate rich diet) following exposure to uremic toxins (exogenously administered in drinking water), was observed. Bioinformatics suggested that exposure to the uremic toxins resulted in a downregulation of pathways associated with stress responsiveness, metabolic activity and calcium-associated processes. Additionally, in the aortas, interestingly, on a short-term exposure (4 days) to IS and pCS, these pathways exhibited an initial downregulation but were predicted to return to normal activity after the end of exposure. In particular, the reactivation of signalling pathway glycoprotein 6 (GP6) was confirmed experimentally after day 6. In conclusion, the authors suggest that the protein-bound microbially produced uremic toxins IS and pCS may be considered promoters for insulin resistance and vascular dysfunction in CKD patients. 

Calciprotein particles (CPP) are clinically connected to inflammation, vascular dysfunction and mortality, mediated by the uremia-dependent transition of amorphous CPP (CPP-I) to the crystalline state (CPP-II); however, the physiology behind this transition (called CPP ripening) is to a large extent not characterized. In CKD, CPP molecules have been implicated in phosphate-induced toxicity. Smith et al. [51] performed detailed physiochemical and biochemical comparisons of endogenous CPP isolated from uremic serum with synthetic CPP (chemically synthesized from 40% of same uremic serum samples), to understand the role of CPP in CKD and eventually CVD. Patients with ESRD undergoing conventional maintenance hemodialysis were included in this study. The bioinformatic analysis revealed that the endogenous CPP comprises a mixture of particles from the synthetic CPP-I and CPP-II. The endogenous CPP also contain bacterial toxins (LPS), microbe-derived components (PGN) and nucleic acids (fragments of RNA, DNA). Based on the protein analysis, cell death and pro-inflammatory process pathways were predicted to be activated by both the endogenous CPP and synthetic CPP-II in a similar pattern. Therefore, the analysis suggests that synthetic CPP-II may be used in place of endogenous CPP as “in vitro” equivalents. 

### 3.3. Proteomics in CKD Progression 

Sepsis-induced acute kidney injury (S-AKI) is often the cause of CKD progression to kidney failure. Lin et al. [52] studied the chronological progression of S-AKI to CKD at a span of seven days in animal models at the kidney proteome and phosphoproteome levels. In addition, given earlier evidence supporting cell apoptosis and inflammatory cell infiltration as major cellular processes in the development and progression of AKI [53], Bcl2-associated agonist of cell death protein (Bad) and FAS-associated death domain protein (Fadd) were specifically analyzed. Fadd levels showed a mild increase on day 7 in CLP in comparison to the sham control mice. Consequently, an increase in the expression of tumor necrosis factor-α (TNFα) and Gas dermin D (Gsdmd), cleavage of caspase-1, phosphorylation of ASC (Pycard) and ERK convincingly pointed towards activation of pyroptosis, formation of inflammasomes and a biologically delayed response of transitioning from S-AKI to CKD. However, the extensive inflammation and histopathology data indicate putative colonization of the kidney by bacteria that crossed the gut barrier. Further studies are needed in order to elucidate the role of gut bacteria in the onset of AKI and subsequently the progression to CKD. For better representation, main findings from the above studies are summarized in Table 2.

## 4. Metabolomics in the Gut-Kidney-Axis

Kanemitsu et al. [54] developed a chemical isotope labelling-LC-MS/MS method using 2-picolylamine and its isotopologue, aiming to more accurately quantify important microbiota-derived carboxyl-containing metabolites, such as short chain fatty acids (SCFAs), secondary bile acids and indole-3-acetic acid (IAA), due to their involvement in many biological processes in CKD. The analysis was performed in plasma, fecal and cecal samples of animal models (with renal failure (RF) and/or GF or SPF). Hematoxylin and eosin (H&E) and Masson’s trichrome staining indicated more severe renal inflammation and reduced severe renal fibrosis in the GF-RF mice compared to the SPF-RF mice. It was also observed that the important metabolites for endothelial barrier function like SCFAs, IAA and n-3 type of polyunsaturated fatty acid were present at significantly lower levels in the GF-RF mice versus the other groups. As reported by the authors, a limitation in this study is that the observed differences in the plasma metabolites between SPF and SPF-RF (or SPF-RF and GF-RF) may be attributed not only to the decline of renal clearance but also to confounding factors such as body weight, body fluid balance, cage effects, and dietary intake, emphasizing the need for adjusting for such potential confounders when investigating the gut-kidney axis. Also, the disruption of the gastrointestinal epithelial barrier function might promote adenine absorption that results in CKD progression, underscoring collectively the need for further studies on the presented findings.

Targeting the characterization of links between the gut microbiome and serum metabolites in the context of CKD, Feng et al. [55] performed an untargeted metabolomics study in rat models. A marked decline in microbial diversity and significant changes in 291 serum metabolites corresponding to lipids, amino acids, bile acids and polyamines were observed in the CKD rats versus controls. Interestingly, creatinine clearance (CCr) was associated with the polyamine metabolism, whereas the systolic blood pressure (SBP) levels were related to the glycine-conjugated metabolites in the CKD rats. Based on these results, administration of poricoic acid A (PAA) and *Poria cocos* (PC) was tested in the rat models, which was found to amend microbial dysbiosis and mitigate hypertension and renal fibrosis. In brief, treatments with PAA and PC lowered the serum levels of microbial-derived products including glycine-conjugated compounds and polyamine metabolites, respectively. Both (PAA and PC) upregulated the protein expression of Zonula occludens protein 1 (ZO1), occludin and claudin-1, inhibited the inflammatory IκB/NF-κB pathway as well as the upregulated cytoprotective (anti-oxidant response regulator) Keap1/Nrf2 pathway in CKD treated (with PAA and PC) rats versus CKD rats. 16S rRNA sequencing suggested the enrichment of *Enterobacteriaceae, Sutterellaceae* and *Clostridiaceae_1* as well as the depletion of *Clostridiaceae_2* and *Leuconostocaceae* in CKD rats versus sham controls. Collectively, the study provided strong links between CKD, gut microbial dysbiosis and metabolic changes in the host, suggesting potential therapeutic dietary strategies to prevent CKD progression. Further investigation especially of the transferability of these results to humans is pending. 

Along the same lines, but focusing on human samples, Wang et al. [56] investigated the links between intestinal microbiome composition, uremic toxins and human ESRD. Blood serum and fecal samples from a total of 223 ESRD patients and 69 healthy controls were collected for this study. The serum metabolome, in particular, was characterized by enrichment of nine gut microbial derived uremic toxins in ESRD versus controls: IS, pCS, trimethylamine-N-oxide (TMAO), phenylacetylglutamine (PAGln), hippuric acid, phenyl sulfate (PS), p-cresyl glucuronide, cinnamoylglycine and phenylacetylglycine (PAGly), along with bile acid compositional imbalances. Similarly, the fecal metabolome showed an enrichment of uremic toxin precursors and secondary bile acids, proposing that in ESRD patients, the alteration of metabolites in intestine and uremic toxin accumulation in serum go hand-in-hand. These metabolic changes paralleled changes in the gut microbiome, with 51% (457 out of 900) of the species exhibiting significant alterations in the ESRD patients versus controls. Species with the highest enrichment in the ESRD patients included *Eggerthella lenta*, *Flavonifractor* spp. (mainly *F. plautii*), *Alistipes* spp. (mainly *A. finegoldii* and *A. shahii*), *Ruminococcus* spp. and *Fusobacterium* spp.; whereas species such as *Prevotella* spp. (mainly *P. copri*), *Clostridium* spp. and several butyrate producers (*Roseburia* spp., *Faecalibacterium prausnitzii* and *Eubacterium rectale*) showed a depletion, suggesting that the observed decrease in short chain fatty acids (SCFAs) in the ESRD patients may be linked to the reduction of SCFA-producing species. To further prove the impact of microbiome on CKD, an additional study was performed involving transferring of fresh microbiota (fecal microbiota transplantation) from ESRD patients into GF mice with induced CKD. This resulted in a higher production of serum uremic toxins and aggravation of renal fibrosis and oxidative stress in the GF-ESRD mice in comparison to the non-transplanted controls. In a parallel similar experiment using rat models for CKD, the species *Eggerthella lenta* and *Fusobacterium nucleatum* were found to increase the production of uremic toxins, which could be reversed following the administration of the probiotic *Bifidobacterium animalis*. This study, therefore, collectively enhanced the hypothesis that certain bacterial species affect the levels of uremic toxins and kidney function in both human and animal models. 

A similar approach was carried out by Wu et al. [32] to determine the relationship between the gut microbiome and circulating host metabolites. In brief, a total of 72 patients were grouped into mild (stage 1 and 2, *n* = 26), moderate (stage 3, *n* = 26) and advanced (stage 4 and 5, *n* = 20) CKD and compared to 20 healthy controls for their blood serum and fecal sample targeted analysis of serum bile acids, SCFAs, medium-chain fatty acids (MCFAs) and uremic toxins (pCS and IS). The authors identified that when progressing from early to advanced stages, *Prevotella sp. 885*, *Weissella confuse*, *Roseburia faecis*, and *Bacteroides eggerthii* levels were significantly reduced, while *Alloscardovia omnicolens*, *Merdibacter massiliensis*, and *Clostridium glycyrrhizinilyticum* levels were elevated significantly. Several species, on the other hand, showed an alteration only in specific stage(s), including *Cetobacterium somerae* (mild CKD), *Candidatus Stoquefichus sp. KLE1796* (mild CKD), *Fusobacterium mortiferum* (moderate CKD), *Bariatricus massiliensis* (moderate CKD), *Bacteroides stercorirosoris* (moderate CKD), and *Merdimonas faecis* (advanced CKD). The six deregulated metabolites across early to late disease were IS, pCS, propionic acid, caproic acid, hepatonic acid and capric acid. These results collectively enhance the importance of host-microbiome-metabolite axis in CKD progression, opening up the way for further studies investigating, closely and at higher power, associations with specific disease stages.

The same research group [57] more recently performed a study to investigate the impact of a low protein diet (LPD) onto the gut microbiota, host serum metabolites and CKD. A total of 16 CKD-LPD patients, 27 CKD patients fed with normal protein diet (CKD-NPD) and 34 healthy controls were selected for the study. The relative abundance of groups *Anaerostipes* and *Eubacterium hallii* increased, while that of groups *Calditerricola*, *Streptococcus anginosus*, *Lactobacillus mucosa* and *Clostridium paraputrificum* significantly decreased in the CKD-LPD versus the CKD-NPD samples. The metabolomics analysis indicated significant increases in the secondary bile acid glyco λ-muricholic acid, significant decreases in the fatty acid nonanoic acid and no difference in the uremic toxins IS and pCS in CKD-LPD versus NPD patients, whereas, when CKD patients were compared to healthy controls, an enrichment of bacterial metabolites related to D-alanine, ketone bodies and glutathione metabolism was observed. It was also suggested that the reduction of serum SCFA levels (acetic, heptanoic and nonanoic acid) in CKD-LPD patients could be associated to the decreased abundance of *Lachnospiraceae* and *Bacteroidaceae* families, considered responsible for the butyrate production. This metabolomics study further suggests the ability of the gut microbiome to adapt to dietary restrictions in renal patients, with potential impacts on the circulating metabolites requiring further study.

The nitrogen-free precursor of an essential amino acid, α-ketoacid, is clinically prescribed to ESRD patients in supplement to LPD. Nevertheless, the direct effect of α-ketoacid on the gut microbiota in CKD has not been investigated. The metabolomic study conducted by Yenan Mo et al. [58] focused on understanding the effect of α-ketoacid administration on the gut microbiota and serum metabolic profile in an adenine-induced CKD rat model. A significant decrease in the levels of serum creatinine, blood urea nitrogen (BUN) and proteinuria with a significant improvement in tubular atrophy, glomerulosclerosis and gut fibrosis were observed when comparing α-ketoacid-CKD versus CKD rats. A significant increase in the abundance of genes *Methanobrevibacter, Akkermansia, Blautia* and *Anaerositipes* and a significant decrease in the abundance of genes *Anaerovorax* and *Coprococcus_3* following treatment with α-Ketoacid of the CKD group was supported by the metabolomic analyses. A significant depletion in the levels of metabolites IS, betaine, choline and cholesterol and an increase in the metabolites PAGly, PAGln and pCS following treatment with α-Ketoacid were observed. A positive correlation (by Spearman’s correlation analysis) could also be detected between the abundance of *Coprococcus_3* with betaine, TMAO, IS, cholic acid and deoxycholic acid serum levels. The authors finally suggest that α-ketoacid exhibits a reno-protective role in the adenine-induced CKD rats, and further studies on its impact on humans could lead to its development as a therapeutic approach in the treatment of CKD.

Trimethylamine-N-oxide (TMAO) is produced by the gut microbiota from trimethylamine obtained from dietary phosphatidylcholine or carnitine and has been associated with CVD and renal dysfunction [59,60]. Nanto-Hara et al. [61] investigated the use of the laxative linaclotide, a guanylate cyclase C agonist, as a novel strategy for the reduction of TMAO levels and prevention of CVD and CKD in animal models (5/6 nephrectomy with or without linaclotide). These analyses revealed that at a low dosage of 10 µg/kg, linaclotide decreased the TMAO plasma levels and at a higher dosage of 100 µg/kg, linaclotide significantly decreased additional uremic toxins (BUN, CCr, urea, trans-aconitate, TMAO, IS, hippurate). The microbiome analysis suggested possible involvement of *Clostridiales* spp. in the observed reduction of the TMAO levels. A quantitative PCR analysis revealed significantly decreased expression of collagen I, TGF-β, Galectin-3 (Gal-3) and ST2 genes on linaclotide treatment which correlated to the reduced levels of Gal-3 and ST2 in the plasma samples. In CKD mice, the inflammatory F4/80-positive macrophages exhibited a high abundance in the small intestine, whereas low levels of colonic claudin-1 were observed. However, these observations were reversed following administration of linaclotide. Therefore, collectively, the study suggested that linaclotide may reduce uremic toxin levels including TMAO, with an important potential therapeutic impact on CKD and its complications, meriting further investigation. 

Traditional Chinese medicines such as Rehmanniae Radix Preparata (RR) and Corni Fructus (CF) are clinically used in the treatment of CKD. In the study performed by Zhang et al. [62] the impact of these drugs administrated individually (RR, CF) or in combination (RC) on the microbiome and host metabolites was investigated. In the case of RC, there was a significant increase in the number of beneficial bacteria such as *Ruminococcaceae UCG-014, Ruminococcus 1, Prevotellaceae_NK3B31_group, Lachnospiraceae NK4A136 group* and *Lachnospiraceae UCG-001* and a significant decrease in the levels of detrimental bacteria *Desulfovibrio*. The metabolomics analysis suggested CKD was associated with deregulation of 15 metabolites that were involved in the amino acid, bile acids and glycerophospholipid metabolism pathways, for example: indole-3-propionic acid (IPA) and phenylpyruvate were downregulated and tryptophan was upregulated. Collectively, this study highlights the vital role of gut microbiota in CKD and its potential as a novel therapeutic target. 

Saggi et al. [63] performed a metabolomics study to investigate the induced metabolic changes and potential benefits of treatment with the probiotic Renadyl™ in 24 CKD patients (Stage III and IV at baseline and after four months of probiotic administration was performed). A total of 14 metabolites were associated with the decreased BUN group, out of which 11 (including betaine, creatine, lipoproteins, lactate, trimethylamine) were considered to be modulated by gut microflora based on existing evidence. These metabolites participated in carbohydrate and choline metabolism and in energy regulation pathways. Further studies on the potential use of such metabolites to monitor benefits from the probiotic administration are pending. 

The impact of microbiome-produced uremic toxins on the kidney is thought to be mediated by specific transporters. In an effort to characterize the mechanism of action of the renal organic anion transporters OAT1 (SLC22A6) and OAT3 (SLC22A8), involved in proximal tubular secretion of uremic toxins, Bush et al. [64] used the rat subtotal nephrectomy model (STN). A plasma metabolomics study quantified 668 metabolites, out of which 58 uremic solutes exhibited higher levels in probenecid (OAT inhibitor)-treated compared to non-treated STN rats. In parallel, the para-aminohippuric acid (PAH) clearance analysis revealed that on a two-hour exposure with probenecid, an inhibition in the OAT(s) mediated tubular secretion of organic anions was observed in the control and STN rat groups. The administration of the inhibitor further proved the central role of these transporters in the tubular secretion of various uremic toxins such as IS, kynurenate and anthranilate. Collectively, the authors concluded that OAT(s) play a vital role in the transport of uremic solutes and toxins in the host tissues and body fluids. 

Focusing on another anion transporter SCLCO4C1, Kikuchi et al. [65] performed an untargeted metabolomics study. Phenyl sulfate (PS) alone showed a significantly higher abundance in the wild type rats versus the transgenic (overexpressing SLCO4C1) rats. Moreover, oral administration of PS was found to elicit podocyte damage and increase glutathione levels with an increase in the expression levels of inflammatory genes: TNF-α and monocyte chemoattractant protein-1 (MCP-1/ Ccl2), the fibrotic gene TGF-α1, fibronectin (Fn1) and collagen I (Col1a1), suggesting a pro-fibrotic and -inflammatory effect of this metabolite. The association of PS with CKD was then further investigated in a prospective study of 362 diabetic patients followed up for a two-year time period. PS levels were found to be predictive of the albumin-to-creatinine (ACR) levels, especially for microalbuminuria in DKD patients. Based on these promising results, the effect of inhibitors (2-aza-tyrosine and L-meta-tyrosine) of the PS-producing tyrosine phenol-lyase (TPL) enzyme on the plasma PS levels, albuminuria, renal function and fecal microbiome composition was subsequently investigated. The analysis revealed that two bacterial orders (*Coriobacteria* and *Erysipelotrichales*) showed a significant increase in the animals with renal failure, and these changes were markedly decreased by 2-aza-tyrosine. Collectively, this study highlighted the role of PS and suggested a potential therapeutic approach for preventing DKD progression via inhibition of the microbial enzyme TPL.

Sun et al. [66] performed a metabolomic study aiming to characterize metabolic changes in CKD with potential links to the gut microbiome. IPA was the only metabolite with significantly reduced levels in the eGFR rapid decline group at baseline and at a one-year follow-up. Interestingly, IPA is an indole-derived tryptophan metabolite produced in the gut by *Clostridium sporogenes* and other bacterial species. The results indicated higher concentrations of IS and pCS in the CKD group compared to the healthy control. In contrast, reduced IPA levels were observed in the CKD patients compared to the heathy controls. The authors’ suggestion that patients with high levels of serum IPA may exhibit a lower risk of rapid decline in their renal function merits further analysis. The findings from the metabolomics studies are summarized in Table 3.

## 5. Expert Opinion

Researching the “leaky gut” theory, scientists aim to restore eubiosis, i.e., the optimal balance of gut microbiota [67]. However, this has proven to be an arduous challenge due to the vast and complex microbial ecosystem [68]. The fact that a single metabolite or microbial species is the cause of multifaceted chronic diseases like CKD is implausible [7]. In order to develop a novel prevention strategy and/or therapeutic treatment for CKD and its associated risk factors such as diabetes, hypertension and vascular calcification (among others), the complex interactions of the gut microbiota with the kidney have to be better characterized and understood. The first step towards this end is to acquire an in-depth interpretation of the uremic-gut microbiome and its associated functions. 

Through this review, we aimed to map the main findings from the application of -omics technologies in the investigation of the gut-kidney axis in CKD. Even though we cannot rule out the possibility that some relevant studies may have not been retrieved, the presented overview, generated based on a systematic search of the literature, may be considered representative of the current status. As shown, the vast majority of retrieved studies focused on the impact of different dietary supplements, such as RS, Saa, LPD, α-ketoacid and/or specific drugs such as probenecid, linaclotide and a combination of RR and CF. These studies involved analyses at the protein, RNA and/or metabolite levels, and their links to overall disease progression using animal models. The results indicate changes in both host and microbiome, including changes in the bacterial produced toxins (such as IS and pCS), metabolic pathways (such as indole metabolism), immune complement system and the AGE pathway. Bacteria related to CKD are involved in butyrate production, mucin degradation and uremic toxins production (TMAO). In some cases, activity of specific bacterial enzymes such as TnaA could be linked to the production of uremic toxins. These experiments, based on animal models, also suggested an added value of such dietary supplements in reducing CKD progression. 

Studies using human samples, as expected, are still limited; nevertheless, the available data support the importance of the host-bacterial interaction for the onset and progression of CKD. In line with the animal model studies, impact of dietary supplements (such as LPD) and /or specific probiotics (such as Renadyl^TM^) on the human metabolic profile, particularly uremic toxin release, has been a main area of research. All these animal and human studies commonly demonstrate that dietary modifications can favorably affect the level of gut-produced proteins and metabolites related to CKD. Moreover, it appears that diet interventions do not radically alter the composition of the gut microbiome. Collectively, it is suggested that these dietary supplements or medications improve tubular atrophy, glomerulosclerosis, kidney fibrosis and gut barrier integrity, eventually reducing the risk of kidney injury and CKD progression.

Metabolomics approaches have been more extensively applied than other -omics for the study of the gut-kidney axis, probably due to the well-established critical role of uremic toxins in CKD. Collectively, these studies highlight the importance of the host-microbiome-metabolite network and its potential for developing novel therapeutic targets. *Bacteroides eggerthii*, PS and IPA were proposed as possible biomarkers for prediction of CKD progression. Potential preventive and/or treatment dietary changes, such as use of PAA and PC, and potential new therapeutic targets, such as organic anion transporters OAT1, OAT3 and SLCO4C1, await further investigation to assess their clinical applicability. These organic anion transporters play a vital role in the transport of microbially produced uremic toxins in CKD. Therefore, exploring their role could lead to the elucidation of molecular processes relevant to CKD pathology. 

It can be further noted that most of these metabolomic studies were performed in combination with 16S rRNA gene sequencing in order to characterize the composition of the gut microbiome. However, this methodology has limitations in analytical output and data interpretation, and exhibits poor reproducibility due to chimera generation and variation in the regions of 16S rRNA amplified gene and reference databases [69]. Therefore, a full RNA sequencing using methods like shotgun metagenome sequencing could allow improved characterization of the gut microbiome at the species and strain level compared to 16S rRNA gene sequencing method [70]. 

The variability in the applied methods, tested species and biological samples, in combination with the relatively small number of retrieved relevant studies, did not allow for a statistical analysis of the results of our search. Nevertheless, interestingly in some cases, overlapping results between two (or more) different studies may be observed: Inflammation and its associated molecular features, such as TNF-α, are reported to significantly increase in CKD in comparison to healthy controls; glutathione metabolism, expression of glucose transporters and glycine-conjugated metabolites associated with energy metabolism are significantly reduced in CKD; claudin-1, collagen I and TGF-β were also reported to decrease at the protein and/or mRNA levels with CKD progression, in all cases linked to some extent to microbial changes. Along these lines, *Clostridium sporogenes* producing tryptophan and its metabolites such as IPA has also been associated with CKD, whereas significant decrease in butyrate-producing bacterial families was observed in CKD models. 

Collectively, it is conclusive by now that the decrease of butyrate-producing bacteria along with an increase in tryptophanase activity and indole synthesis can be directly related to the increased inflammation and concentrations of uremic toxins (IS) in CKD. Tryptophan is an essential amino acid (for humans and bacteria), and its usual intake is by dietary proteins. The gut bacteria take up tryptophan and convert it to indole through the action of the enzyme TnaA. Indole enters the blood circulation and is modified in the liver to Indoxyl-sulfate, a uremic toxin [71]. Therefore, a low protein diet for CKD patients is usually recommended and, conversely, a high tryptophan intake is deleterious for CKD patients, since it increases levels of uremic toxins and promotes inflammation. Therefore, these pathways and metabolites consistently reported across various research studies could, in fact, form the base of future research in treatment and/or prevention of CKD with a gut microbiome perspective.

## 6. Future Perspective

The knowledge of combining disease-associated microbiome data with the application of modern DNA technologies paves a path for the emerging smart “engineered” gut microbiome [72]. The beneficial modification of the gut microbiome could be achieved by medications or dietary interventions [14]. Integrating such information on gut microbiome with -omics data reflecting the gut-kidney interactions can open up the way to more effective personalized treatment approaches. As shown through this review, multiple disparate data exist, yet a spherical simultaneous analysis at multiple -omics levels in both host and microbiome in association to disease is still lacking. Aiming to fill this gap as much as possible, the “STRATEGY-CKD” research and training programme (https://www.strategy-ckd.eu/), bringing together scientists from 7 different countries in Europe and supported financially by the European Union, aims to develop an integrative analysis of the gut-kidney axis with therapeutic implications. Further efforts in this direction are no doubt required. Even though animal models can undoubtedly assist in investigating selected therapeutic approaches and biological mechanisms, defining microbiome associations to human disease requires well-designed human studies. The vast inter-individual variability of microbiomes restricts the power of individual observations, rendering the need for an open-minded data sharing approach even more vital. This is required to facilitate data exchange and cross-correlation towards increasing the power of individual observations. An additional factor that emerges is the need for recording dietary intake and supplements, these comprising important confounders when investigating microbiome impact and defining biological associations with disease. An effort of the community to generate a list of common data elements to be employed in gut-kidney investigations appears relevant and of anticipated added value, facilitating data comparability and integration and thus, ultimately, increasing coverage and breadth of our knowledge on the involvement of the gut microbiome in CKD.

## Figures and Tables

**Figure 1 toxins-14-00176-f001:**
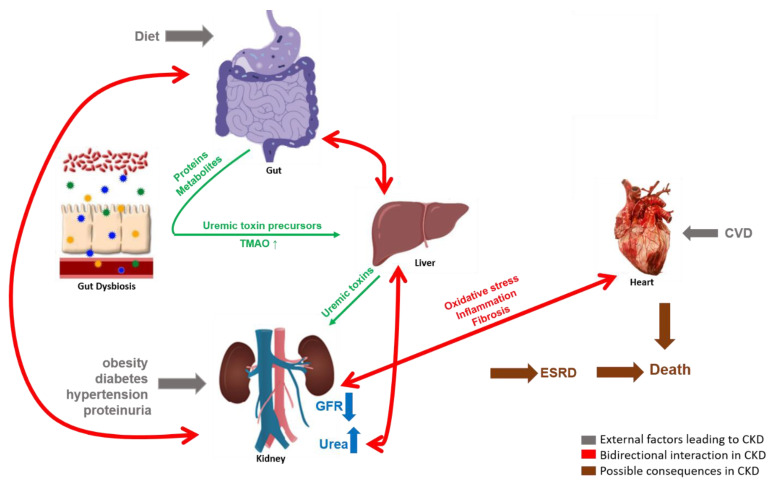
The complex bidirectional interactions of the gut-kidney axis. Gut microbiota produce proteins and metabolites acting as uremic toxin precursors which are converted into uremic toxins in the liver and are normally excreted by the kidneys in urine. External factors such as diet, obesity, diabetes, hypertension, proteinuria and cardiovascular disease (CVD) cause CKD, as measured by a reduced GFR and/or increased urea concentrations. In the case of CKD, the microbially produced uremic toxins accumulate in the host system, leading to enhanced oxidative stress, inflammation and/or fibrosis which may have a detrimental effect on the functionality of gut microbiota and other host organs like heart and kidneys, forming a vicious cycle.

**Figure 2 toxins-14-00176-f002:**
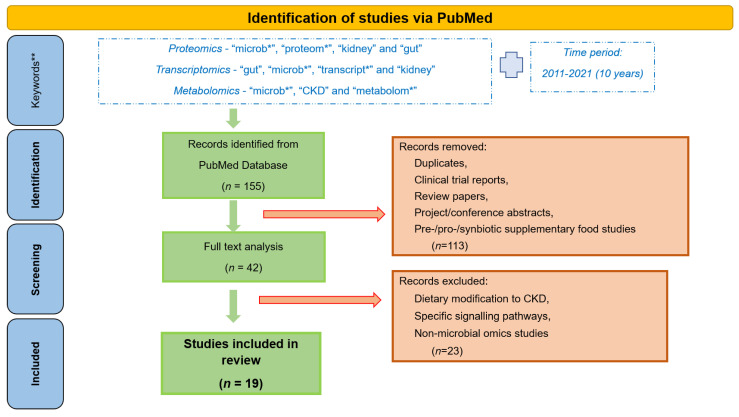
Workflow followed for the retrieval of the presented omics studies associated with gut microbiota and CKD. *—keywords were used with “*” to perform a broader search, irrespective of their use in multiple forms, for example plurals. **—additional keywords used for the search include “renal”, “gut”, “*omics”, etc., but yielded in most extent overlapping results.

**Figure 3 toxins-14-00176-f003:**
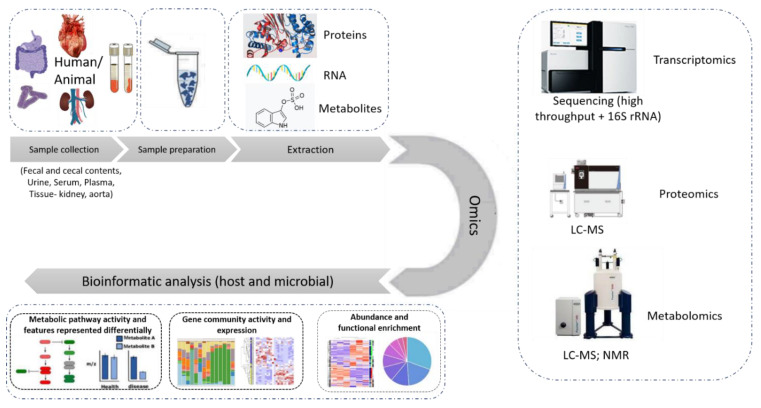
State-of-the-art omics experimental study. The different types of samples used in an omics study of relevance to this review are fecal and cecal contents, serum, plasma, urine, kidney and aorta tissue samples. After the samples are prepared following a chemical protocol in the laboratory, the extracted proteins, RNA or metabolites are subjected to the respective -omics analysis, performed by state-of-the-art instruments like LC-MS, high throughput sequencer or nuclear magnetic resonance (NMR) spectroscopy. The data collected are then subjected to further bioinformatic analysis targeting their integration into molecular pathways and the better understanding of the system.

**Table 1 toxins-14-00176-t001:** Summary of experimental study utilizing Transcriptomics in the gut-kidney axis.

Reference	Organism	Design and Sample	Key Findings
Snelson [35]	Mice	HT* + RS* (*n* = 5) Vs C* (*n* = 5); HT (*n* = 5) in RNA extracted from Gut sections and Kidney cortex	systemic innate immune complement system: C3* and C5* effector molecules ↓ alagebrium inhibited the AGE pathway gut barrier integrity ↑ risk of kidney injury ↓

***Abbreviations**—**HT**: heat treated, **RS**: resistant starch, **C**: healthy control, **AGE**: advanced glycation pathway, **C3 and C5**: complement system end effector molecules.

**Table 2 toxins-14-00176-t002:** Summary of experimental studies utilizing Proteomics in the gut-kidney axis.

Reference	Organism	Design and Sample	Key Findings	Validation
Zybailov [42]	Rats	CKD-DS* (*n* = 9) Vs CKD-RS* (*n* = 9) in Frozen cecal content	Thioredoxin, S100-A6* ↑ serine-type endopeptidase inhibitor and metalloexopeptidase activities ↑ fibrolytic *Ruminoccocus* ↑ amylolytic *Ruminoccocus* ↓	-
Karaduta [44]	Mice	CKD (*n* = 4); CKD-RS (*n* = 4) Vs H* (*n* = 4); H-RS (*n* = 4) in Frozen cecal content	In CKD-RS: Indole mechanism ↓ butyrate-producing bacteria ↑ mucin-degrading bacteria ↓ IS* ↓ bacterial proliferation ↑	-
Lobel [49]	Mice	high-Saa* diet Vs low-Saa diet in Cecal content	Spp1*, Tgfb1*, Icam1*, Ccl2*, Timp1* ↓ Indole mechanism ↓ *Escherichia coli (E. coli)* ↑ tryptophanase (TnaA) ↓	-
Opdebeeck [50]	Rats	IS (*n* = 14) or pCS* (*n* = 14) Vs Vehicle (*n* = 14) in Aorta and Kidney tissue	In aorta: TGF-β and collagen I ↓ in IS GLUT1* ↓ in pCS No change in IL-1β*, klotho, TNF-α* and vascular cell adhesion molecule 1 → stress responsiveness pathway ↓ metabolic activity pathway ↓ calcium-associated processes ↓ → Glycoprotein 6 (GP6) reactivated on end of exposure	RT-PCR: GLUT1 -aorta. IL-1β, collagen I, klotho, TGF-β, TNF-α, and vascular cell adhesion molecule 1-left kidney.
Smith [51]	Human	Endogenous CPP* Vs Synthetic CPP-I & -II in Serum samples	Cell death pathways and pro-inflammatory process pathways activated in same way use of synthetic CPP-II in place of endogenous CPP as “in vitro” equivalents	-
Lin [52]	Mice	S-AKI* to CKD progression at day 2 and 7 in Kidney tissue (*n* = 18)	Hmgcs2*, S100-A8, Chil3* ↑ TNFα, Gsdmd*, caspase-1, ASC and ERK ↑ deregulation of mitochondrial inner membrane proteins Atp5j*, Ndufb1*, Cox2*	Immunoblot: Hmgcs2, S100-A8, Chil3, TNFα, Gsdmd, caspase-1

***Abbreviations**—**DS**: digestible starch, **RS**: resistant starch, **H**: healthy control, **S100-A6**: calcylin, **IS**: indoxyl sulfate, **Saa+Ade**: sulphur containing amino acid + adenine, **Spp1**: osteopontin, **TGF**: transforming growth factor, **b1**: beta-1 proprotein, **Icam1****:** intercellular adhesion molecule 1, **Ccl2**: C-C motif chemokine 2, **Timp1**: metalloproteinase inhibitor 1, **pCS**: p-cresyl sulfate, **GLUT1:** glucose transporter 1, **IL-1β**: Interleukin-1β, **TNF-α**: tumor necrosis factor-α, **CPP**: calciprotein particles, **S-AKI**: sepsis-induced acute kidney injury, **Hmgcs2**: Hydroxymethylglutaryl-CoA synthase, **Chil3**: chitinase-like protein 3, **Gsdmd**: Gas dermin D, **Atp5j**: ATP synthase-coupling factor 6, **Ndufb1**: NADH dehydrogenase [ubiquinone] 1 beta subcomplex subunit 1, **Cox2**: cytochrome c oxidase like Cyclooxygenase-2.

**Table 3 toxins-14-00176-t003:** Summary of experimental studies utilizing Metabolomics in the gut-kidney axis.

Reference	Organism	Design and Sample	Key Findings	Validation
Kanemitsu [54]	Mice	GF*-RF* (*n* = 5); SPF*-RF (*n* = 3) Vs GF (*n* = 4); SPF (*n* = 4) in Plasma, Fecal and Cecal contents	In GF-RF: renal inflammation ↑ renal fibrosis ↓ SCFAs*, IAA* ↓ n-3 type of polyunsaturated fatty acid ↓	-
Feng [55]	Rats	Study 1: CKD (*n* = 6) Vs Control (*n* = 6) in Serum, Colonic luminal contents, Colon tissues and Kidney tissues	In CKD: CCr* associated with polyamine metabolism & SBP* with glycine-conjugated metabolites.	
	Rats	Study 2: PAA-CKD (*n* = 8); PC-CKD (*n* = 8) Vs Controls (*n* = 8) in Serum, Colonic luminal contents, Colon tissues and Kidney tissues	In poricoic acid A (PAA) and Poria cocos (PC): microbial dysbiosis, hypertension and renal fibrosis ↓ ZO1*, occludin and claudin-1 ↑ IκB/NF-κB pathway ↓ Keap1/Nrf2 pathway ↑ *Enterobacteriaceae, Sutterellaceae* and *Clostridiaceae_1* ↑ *Clostridiaceae_2* and *Leuconostocaceae* ↓	Western blot: ZO1, occludin and claudin-1
Wang [56]	Human	Study 1: ESRD* (*n* = 223) Vs Healthy controls (*n* = 69) in Serum and Fecal samples	microbial derived uremic toxins ↑ uremic toxin precursors and secondary bile acids ↑ *Eggerthella lenta*, *Flavonifractor* spp. (mainly *F. plautii*), *Alistipes* spp. (mainly *A. finegoldii* and *A. shahii*), *Ruminococcus* spp. And *Fusobacterium* spp. ↑ *Prevotella* spp. (mainly *P. copri*), *Clostridium* spp. And several butyrate producers (*Roseburia* spp., *Faecalibacterium prausnitzii* and *Eubacterium rectale*) ↓	
	Mice	Study 2: Transplantation of fecal metabolome from ESRD patients to GF-mice (*n* = 13) Vs Controls (*n* = 13)	In GF-ESRD mice: serum uremic toxins ↑ renal fibrosis and oxidative stress ↑	Fecal microbiota transplantation from patient into GF-mice: serum uremic toxins ↑ renal fibrosis and oxidative stress ↑
	Rats	Study 3: Transplantation of fecal metabolome from ESRD patients to antibiotics treated rats (*n* = 9) Vs Controls (*n* = 9) in Serum and Fecal samples	In rats: *Eggerthella lenta* and *Fusobacterium nucleatum* increased production of uremic toxins *Bifidobacterium animali* reduced uremic toxins production	
Wu [32]	Human	CKD mild (stage 1 and 2, *n* = 26), moderate (stage 3, *n* = 26) and advanced (stage 4 and 5, *n* = 20) Vs Healthy controls (*n* = 20) in Fecal and Serum samples	*Prevotella sp. 885*, *Weissella confuse*, *Roseburia faecis*, and *Bacteroides eggerthii ↓* *Alloscardovia omnicolens*, *Merdibacter massiliensis*, and *Clostridium glycyrrhizinilyticum ↑* IS*, pCS*, hepatonic acid ↑ propionic acid, caproic acid ↓	-
Wu [57]	Human	CKD-LPD* (*n* = 16) Vs Healthy controls (*n* = 34); CKD-NPD* (*n* = 27) in Fecal and Serum samples	*Anaerostipes* and *Eubacterium hallii ↑* *Calditerricola*, *Streptococcus anginosus*, *Lactobacillus mucosa* and *Clostridium paraputrificum ↓* glyco λ-muricholic acid ↑ nonanoic acid ↓ D-alanine, ketone bodies and glutathione metabolism metabolites ↑ Butyrate producing bacteria (*Lachnospiraceae* and *Bacteroidaceae* families) ↓	-
Yenan Mo [58]	Rats	α-ketoacid + CKD (*n* = 8) Vs Control (*n* = 8); CKD (*n* = 8) in Fecal and Serum samples	tubular atrophy, glomerulosclerosis and gut fibrosis ↑ *Methanobrevibacter, Akkermansia, Blautia* and *Anaerositipes ↑* *Anaerovorax* and *Coprococcus_3 ↓* metabolites IS, betaine, choline and cholesterol ↓ metabolites PAGly*, PAGln*, and pCS ↑ → α-ketoacid exhibits a reno-protective in CKD rats	-
Nanto-Hara [61]	Mice	CKD-linaclotide (*n* = 6) Vs CKD (*n* = 5) in Kidney tissue, Fecal and Plasma samples	collagen I, TGF-β*, Galectin-3 (Gal-3) and ST2 genes ↓ F4/80-positive macrophages in small intestine ↑ colonic claudin-1 ↓ → low dosage of linaclotide (10 µg/kg) decreased TMAO* → higher dosage of linaclotide (100 µg/kg) decreased BUN*, CCr, urea, trans-aconitate, TMAO, IS and Hippurate	qPCR: collagen I, TGF-β, Galectin-3 (Gal-3) and ST2 genes
Zhang [62]	Rats	CKD-RC* (*n* = 6) Vs Normal (*n* = 6); CKD (*n* = 6); CKD-RR* (*n* = 6); CKD-CF* (*n* = 6) in Fecal samples	*Ruminococcaceae UCG-014, Ruminococcus 1, Prevotellaceae_NK3B31_group, Lachnospiraceae NK4A136 group* and *Lachnospiraceae UCG-001 ↑* *Desulfovibrio ↓* indole-3-propionic acid and phenylpyruvate ↓ tryptophan *↑*	-
Saggi [63]	Human	CKD patients administered probiotic Renadyl™ (*n* = 24) in Plasma samples	16 patients BUN ↓ 11 gut-modulated metabolites found (including betaine, creatine, lipoproteins, lactate and trimethylamine) in decreased BUN patient plasma *Pathways involved:* carbohydrate and choline metabolism energy metabolism and regulation	-
Bush [64]	Rats	Probenecid STN* (*n* = 5) Vs Probenecid control (*n* = 5) Vs STN (*n* = 5) in Plasma samples	*In probenecid treated STN rats:* 58 uremic toxins ↑ *In probenecid treated control and STN rats:* OAT(s)* mediated tubular secretion of organic anions ↓ → central role of OAT(s) transporters in the tubular secretion of various uremic toxins (IS, kynurenate and anthranilate)	-
Kikuchi [65]	Rats	Study 1: Wild type DKD* (*n* = 4) Vs Transgenic overexpressing SLCO4C1* DKD (*n* = 4) in Plasma and Fecal samples	In wild type DKD (non-transgenic) rats: phenyl sulfate (PS) ↑	
	Mice	Study 2: PS + db/db* mice (*n* = 6) Vs Control db/db mice (*n* = 5) in Plasma and Fecal samples	In PS administered db/db mice: TNF*-α, MCP-1* (Ccl2*), TGF-α1, Fn1* and collagen I (Col1a1) ↑ → elicit podocyte damage and increase glutathione levels	qPCR: TNF-α, MCP-1 (Ccl2), TGF-α1, Fn1 and collagen I (Col1a1)
	Human	Study 3: DKD patients at t = 0 (*n* = 362) Vs DKD patients at t = 2 years (*n* = 362) in Plasma and Fecal samples	→ PS predictive of the ACR levels, especially for microalbuminuria in DKD patients.	
	Mice	Study 4: Pre-treatment** db/db mice (*n* = 10) Vs After-treatment ** db/db mice (*n* = 10) in Plasma and Fecal samples	On treatment with **2-aza-tyrosine: *Coriobacteria* and *Erysipelotrichales* ↓	
Sun [66]	Human	Study 1: eGFRRD* patient (*n* = 10) Vs Healthy control (*n* = 10) at t = 0 and t = 1 year in Serum samples	In eGFRRD patients: IPA* ↓ at t = 0 and t = 1 year	-
	Human	Study 2: CKD patients (*n* = 140) Vs Healthy controls (*n* = 144) in Serum samples	In CKD patients: IS and pCS ↑ IPA ↓ → high levels of serum IPA may indicate a lower risk of rapid decline in their renal function and developing CKD.	

***Abbeviations**—**GF**: germ-free, **RF**: renal failure, **SPF**: specific pathogen free, **SCFAs**: short chain fatty acids, **IAA**: indole-3-acetic acid, **CCr**: creatinine clearance, **SBP**: systolic blood pressure, **ZO1**: Zonula occludens protein 1, **ESRD**: end stage renal disease, **IS**: indoxyl sulfate, **pCS**: p-cresyl sulfate, **LPD**: low protein diet, **NPD**: normal protein diet, **PAGly**: phenylacetylglycine, **PAGln**: phenylacetylglutamine, **TGF**: transforming growth factor, **TMAO**: Trimethylamine-N-oxide, **qPCR**: quantitative polymerase chain reaction, **RR**: Rehmanniae Radix Preparata, **CF**: Corni Fructus, **RC**: RR+CF, **STN**: subtotal nephrectomy model rats, **OAT**(**s**): organic anion transporters, **DKD**: diabetic kidney disease, **TNF**: tumor necrosis factor, **MCP-1**: Monocyte chemoattractant protein-1, **Ccl2**: C-C motif chemokine 2, **db/db**: diabetic, **Fn1**: fibronectin, **eGFRRD**: rapid decline in eGFR (estimated glomerular filtration rate), **IPA**: indole-3-propionic acid.

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
