# Peer review of "Microbiome in Chronic Kidney Disease (CKD): An Omics Perspective"

_toxins, 2022, doi:10.3390/toxins14030176_

Round 1

Reviewer 1 Report

As I already mentioned, the topic is very significant and there is a need for an informative review of the newest literature that includes omics studies.

The authors made significant changes within the abstract to highlight the main scope of the manuscript.

In the revised version of the manuscript, the authors improved the introduction and discussed limitations as well as future perspectives.

Author Response

“As I already mentioned, the topic is very significant and there is a need for an informative review of the newest literature that includes omics studies. The authors made significant changes within the abstract to highlight the main scope of the manuscript. In the revised version of the manuscript, the authors improved the introduction and discussed limitations as well as future perspectives.”

  • We cordially thank the reviewer very much for acknowledging the efforts made behind the revised manuscript.

Reviewer 2 Report

  1. Publications within ten years are included. Why is ten years a good cutoff point? Are there important ones before 2012?

  1. I searched for “Microbiome CKD” and found some important papers not included in this manuscript.

Ref 1: Sampaio-Maia B, Simões-Silva L, Pestana M, Araujo R, Soares-Silva IJ. The Role of the Gut Microbiome on Chronic Kidney Disease. Adv Appl Microbiol. 2016;96:65-94. doi: 10.1016/bs.aambs.2016.06.002. Epub 2016 Jul 18. PMID: 27565581.

Ref 2: Clinical Microbiology, Research Article, 12 January 2021. Associations of Genetic Variants Contributing to Gut Microbiota Composition in Immunoglobin A Nephropathy. Authors: Jia-Wei He https://orcid.org/0000-0003-1636-1823, Xu-Jie Zhou https://orcid.org/0000-0002-7215-707X, Ya-Feng Li, Yan-Na Wang, Li-Jun Liu, Su-Fang Shi, Xiao-Hong Xin, Rong-Shan Li, Mario Falchi, Ji-Cheng Lv, Hong Zhang

Therefore, I worry that 19 papers discussed in this review may be unsatisfactory. Perhaps more articles should be included, and other search engines should be used, such as google.

Author Response

“1. Publications within ten years are included. Why is ten years a good cutoff point? Are there important ones before 2012?”

  • Thank you for your detailed review report. This is indeed a very well taken point.
  • The focus of this manuscript was to bring together the “-omics” methodologies and technologies (employed in better understanding of the axis gut-kidney).
  • As a result, it was important to include only the latest advancements and current techniques employed in this field.
  • We believe that with the constantly developing technologies and the evolved applications from MALDI-TOF MS to LC-MS/MS, the cut-off point of 10 years was apt for this kind of a review.

 “2. I searched for “Microbiome CKD” and found some important papers not included in this manuscript.

Ref 1: Sampaio-Maia B, Simões-Silva L, Pestana M, Araujo R, Soares-Silva IJ. The Role of the Gut Microbiome on Chronic Kidney Disease. Adv Appl Microbiol. 2016;96:65-94. doi: 10.1016/bs.aambs.2016.06.002. Epub 2016 Jul 18. PMID: 27565581.”

  • This article mentioned in the Ref 1 is a review paper; we have focused on original papers in our systematic review: ‘Studies focusing on a particular dietary modification for CKD patients or specific signalling pathways, pre-/pro-/syn-biotic supplement studies, non-microbial omics studies, review papers, clinical trial reports, project and conference abstracts were excluded, as represented in Figure 2. (Line 116)’
  • However, we agree that this was an important article explaining the axis gut-kidney and therefore, we have added it as a reference in the introduction section (Line 59 and Reference number 24).  

“Ref 2: Clinical Microbiology, Research Article, 12 January 2021. Associations of Genetic Variants Contributing to Gut Microbiota Composition in Immunoglobin A Nephropathy. Authors: Jia-Wei He https://orcid.org/0000-0003-1636-1823, Xu-Jie Zhou https://orcid.org/0000-0002-7215-707X, Ya-Feng Li, Yan-Na Wang, Li-Jun Liu, Su-Fang Shi, Xiao-Hong Xin, Rong-Shan Li, Mario Falchi, Ji-Cheng Lv, Hong Zhang”

  • This original article is on genomic alterations, whereas, our manuscript focused exclusively on transcriptomics, proteomics, metabolomics as stated in the abstract: ‘Here, we present the latest omics (transcriptomics, proteomics and metabolomics) studies that explore the connection between CKD and gut microbiome (Line 8).’
  • In addition, we acknowledge that research papers may have been accidentally omitted based on the key words used as kindly mentioned in the manuscript: ‘Even though we cannot rule out that some relevant studies may have not been retrieved, the presented overview, generated based on a systematic search of the literature, may still be considered representative of the current status. (Line 502)’

“Therefore, I worry that 19 papers discussed in this review may be unsatisfactory. Perhaps more articles should be included, and other search engines should be used, such as google.”

  • Again, we agree that we may have missed including some relevant research papers. However, we clearly provide to the reader the keywords used in our systematic search. In addition, we strongly believe that PubMed: comprising of more than 33 million peer reviewed citations for biomedical literature, is the most sought-after and trusted search engine when it comes to scientific publishing.

Reviewer 3 Report

The authors have reviewed microbiome in chronic kidney disease (CKD) with several literatures. It is clear the methods to search manuscripts and they reported them in detail. This review article may help readers to understand the recent progress of omics studies focused on CKD progression and -related disease.

  1. In the Tables, several papers are not clear for the material in the Key findings. For example, in Table 1, Ref 46 examined proteome in aorta and kidney tissues while I cannot understand which tissues were related in Key findings, such as TGF-b collagen 1.
  2. In the Table 3, the authors should put the materials
  3. The authors described calciprotein particles, but is this important for microbiome and omics in CKD?
  4. In Figure 1 and 3, the positions of aorta and vena cava are wrong.

Author Response

“The authors have reviewed microbiome in chronic kidney disease (CKD) with several literatures. It is clear the methods to search manuscripts and they reported them in detail. This review article may help readers to understand the recent progress of omics studies focused on CKD progression and -related disease.”

  • We are grateful for this support and appreciation.

“1. In the Tables, several papers are not clear for the material in the Key findings. For example, in Table 1, Ref 46 examined proteome in aorta and kidney tissues while I cannot understand which tissues were related in Key findings, such as TGF-b collagen 1.

  1. In the Table 3, the authors should put the materials”

- We are grateful that you noticed these omissions; the needed information to clarify these points is now provided as follows: 

- Table 2:  The key findings mentioned in the table are of the proteomic study performed in the aorta samples.

Opdebeeck [47]

Rats

IS (n=14) or pCS* (n=14)

Vs

Vehicle (n=14)

in Aorta, Kidney tissue

In aorta:

TGF-β and collagen I ↓ in IS

GLUT1* ↓ in pCS

No change in IL-1β*, klotho, TNF-α* and vascular cell adhesion molecule 1

→ stress responsiveness pathway ↓

metabolic activity pathway ↓

calcium-associated processes ↓

→ Glycoprotein 6 (GP6) reactivated on end of exposure

RT-PCR: GLUT1 -aorta.

IL-1β, collagen I, klotho, TGF-β, TNF-α, and vascular cell adhesion molecule 1 - left kidney.

  • Table 3:

Wang [55]

Human

Mice

Rats

Study 1:

ESRD* (n=223)

Vs

Healthy controls (n=69)

in Serum and Fecal samples

Study 2:

Transplantation of fecal metabolome from ESRD patients to GF-mice (n=13)

Vs

Controls (n=13)

Study 3:

Transplantation of fecal metabolome from ESRD patients to antibiotics treated rats (n=9)

Vs

Controls (n=9) in Serum and Fecal samples

microbial derived uremic toxins ↑

uremic toxin precursors and secondary bile acids ↑

Eggerthella lenta, Flavonifractor spp. (mainly F. plautii), Alistipes spp. (mainly A. finegoldii and A. shahii), Ruminococcus spp. and Fusobacterium spp. ↑

Prevotella spp. (mainly P. copri), Clostridium spp. and several butyrate producers (Roseburia spp., Faecalibacterium prausnitzii and Eubacterium rectale) ↓

In GF-ESRD mice:

serum uremic toxins ↑

renal fibrosis and oxidative stress ↑

In rats:

Eggerthella lenta and Fusobacterium nucleatum increased production of uremic toxins

Bifidobacterium animali reduced uremic toxins production

Fecal microbiota transplantation from patient into GF-mice:

serum uremic toxins ↑

renal fibrosis and oxidative stress ↑

“3. The authors described calciprotein particles, but is this important for microbiome and omics in CKD?”

  • We thank the reviewer for this point. The available data in fact indicate that Calciproteins (CPP) play a crucial role in the heart-gut-kidney axis and impact the progression of CVD and CKD (a bidirectional relationship – Figure 1), accompanied by various other factors such as inflammation and mortality. This is explained in the manuscript: ‘Calciprotein particles (CPP) are clinically connected to inflammation, vascular dysfunction and mortality, mediated by the uraemia-dependent transition of amorphous CPP (CPP-I) to the crystalline state (CPP-II) (Line 237).’
  • In addition, the research study focusing on CPP described in this manuscript, predicts the direct link between CPP and gut microbiota, using proteomics: ‘The endogenous CPP also contain bacterial toxins (LPS), microbe-derived components (PGN) and nucleic acids (fragments of RNA, DNA) … (Line 247).’

“4. In Figure 1 and 3, the positions of aorta and vena cava are wrong.”

  • Thank you for pointing this out. We have modified the picture.

Reviewer 4 Report

In this manuscript, authors summarized the previously reported omics data to demonstrate the association between gut microbiome and CKD, specially highlighted the microbes involved in aromatic amino acid degradation and the decrease of butyrate producing bacterium relative abundance contributing to CKD onset and progression. The transcriptomics, proteomics and metabolomics findings in gut-kidney-axis were presented, which are helpful to develop several therapeutical targets to attenuate CKD progression. The manuscript was well-drafted and organized. One minor suggestion is provided below.

Since transcriptomics controls proteomics, then proteomics controls metabolomics, in Figure 3 right panel, authors should exchange the positions of Proteomics with Transcriptomics. In parallel, the paragraph  "Transcriptomics in the gut-kidney axis" should be presented in front of "Proteomics in the investigation of gut-kidney-axis".

Author Response

“In this manuscript, authors summarized the previously reported omics data to demonstrate the association between gut microbiome and CKD, specially highlighted the microbes involved in aromatic amino acid degradation and the decrease of butyrate producing bacterium relative abundance contributing to CKD onset and progression. The transcriptomics, proteomics and metabolomics findings in gut-kidney-axis were presented, which are helpful to develop several therapeutical targets to attenuate CKD progression. The manuscript was well-drafted and organized.”

  • We greatly appreciate these kind and positive comments.

“One minor suggestion is provided below.

Since transcriptomics controls proteomics, then proteomics controls metabolomics, in Figure 3 right panel, authors should exchange the positions of Proteomics with Transcriptomics. In parallel, the paragraph "Transcriptomics in the gut-kidney axis" should be presented in front of "Proteomics in the investigation of gut-kidney-axis".”

  • We have made the change in order, as suggested.

Round 2

Reviewer 2 Report

No more comments